# Peripheral Nerve Injury Treatments and Advances: One Health Perspective

**DOI:** 10.3390/ijms23020918

**Published:** 2022-01-14

**Authors:** Bruna Lopes, Patrícia Sousa, Rui Alvites, Mariana Branquinho, Ana Catarina Sousa, Carla Mendonça, Luís Miguel Atayde, Ana Lúcia Luís, Artur S. P. Varejão, Ana Colette Maurício

**Affiliations:** 1Centro de Estudos de Ciência Animal (CECA), Instituto de Ciências, Tecnologias e Agroambiente (ICETA) da Universidade do Porto, Praça Gomes Teixeira, Apartado 55142, 4051-401 Porto, Portugal; brunisabel95@gmail.com (B.L.); pfrfs_10@hotmail.com (P.S.); ruialvites@hotmail.com (R.A.); m.esteves.vieira@gmail.com (M.B.); anacatarinasoaressousa@hotmail.com (A.C.S.); cmmendonca@icbas.up.pt (C.M.); ataydelm@gmail.com (L.M.A.); alluis@icbas.up.pt (A.L.L.); 2Departamento de Clínicas Veterinárias, Instituto de Ciências Biomédicas de Abel Salazar (ICBAS), Universidade do Porto (UP), Rua de Jorge Viterbo Ferreira, nº 228, 4050-313 Porto, Portugal; 3Department of Veterinary Sciences, University of Trás-os-Montes e Alto Douro, UTAD, Quinta de Prados, 5000-801 Vila Real, Portugal; avarejao@utad.pt; 4CECAV, Centre for Animal Sciences and Veterinary Studies, University of Trás-os-Montes e Alto Douro, Quinta de Prados, 5000-801 Vila Real, Portugal

**Keywords:** mesenchymal stem cells, nerve guide conduits, nerve recovery, one health, peripheral nerve injury, secretome

## Abstract

Peripheral nerve injuries (PNI) can have several etiologies, such as trauma and iatrogenic interventions, that can lead to the loss of structure and/or function impairment. These changes can cause partial or complete loss of motor and sensory functions, physical disability, and neuropathic pain, which in turn can affect the quality of life. This review aims to revisit the concepts associated with the PNI and the anatomy of the peripheral nerve is detailed to explain the different types of injury. Then, some of the available therapeutic strategies are explained, including surgical methods, pharmacological therapies, and the use of cell-based therapies alone or in combination with biomaterials in the form of tube guides. Nevertheless, even with the various available treatments, it is difficult to achieve a perfect outcome with complete functional recovery. This review aims to enhance the importance of new therapies, especially in severe lesions, to overcome limitations and achieve better outcomes. The urge for new approaches and the understanding of the different methods to evaluate nerve regeneration is fundamental from a One Health perspective. In vitro models followed by in vivo models are very important to be able to translate the achievements to human medicine.

## 1. Introduction

The peripheral nerve system (PNS) is the extensive grid of nerves that functionally integrate diverse parts of the body with the central nervous system (CNS) [1,2]. The PNS results from the combination of the ventral root and the dorsal root both originated from the spinal cord. The first includes the motor neurons and the later the sensory neurons, for which cell bodies reside in the dorsal root ganglion. In the ventral horn of the spinal cord and in specific nuclei of the brainstem are located the cell bodies of the motor neurons. The sensory and motor neuron axons can communicate with distant target organs. In addition, the PNS is more easily regenerated in comparison to the CNS [3] due to their distinct functional environments, integrity of the injured nerve, age of the affected subject, and type of injury sustained. PNI is a neurological condition that seriously threatens both human and animal patients, leading to serious and long-term functional and physiological disabilities [4,5].

Peripheral nerve fibers are very delicate structures, making them easily damaged by different types of compressions [2]. The consequent communicational abnormalities between the brain and the target muscles/organs are the manifestation of the damage [2]. Nerve defects can have multiple origins, such as iatrogenic (medical or surgical) and traumatic [5,6]. A big percentage of PNI is associated with poor functional outcomes, insufficient nerve recovery, and the loss of sensory and motor function. These are followed by partial recovery, muscle atrophy, chronic pain, and profound weakness [7]. Often, axons have to regenerate over long distances at a slow rate of 1–3 mm per day to reinnervate and reach distal motor endplates [8]. For that reason, the regenerative process takes time, especially without any external intervention [4,9]. Additionally, there are other factors that impair nerve regeneration after PNI, such as the loss of extensive amounts of nerve tissue and prolonged denervation of the proximal nerves that enhance the probability of irreversible atrophy of the innervated organs [4].

The types of nerve injuries are classified based on the severity and extent of the damage, creating different outcome possibilities and the recovery depends on the severity [5,10]. To promote better outcomes, it is important to guarantee a rapid intervention. However, even doing so, the prolonged denervation of the nerve segments can lead to low recovery rates and to other disabilities [5,10].

Treatments are mainly microsurgical interventions either with direct repair, tension free end-to-end suture, and—the gold standard technique—the use of autologous nerve grafts to repair larger gaps [6,11,12]. The use of the latter reduces the risk of immunological rejection and provides a native tissue microenvironment that triggers a positive therapeutic effect [4,7,9]. However, it has some limitations, such as donor site morbidity, lack of donor sources, sensory loss, scarring, and neuroma formation [4,7,9]. Recently, many studies have focused on establishing new methods to promote axonal regeneration, ideally without sacrificing other healthy functioning nerves, and improving the PNI outcome. Nerve guide conduits (NGCs) are a viable alternative to autologous nerve grafting [9]. The use of Schwann cells (SCs) or mesenchymal stem cells (MSCs) in synthetic permeable nerve conduits is one of the focuses for an ideal treatment [4]. The animal model has the advantage of having multimodal approaches and different combinations of methods to study axonal regeneration. In addition, in the One Health context, assessment of nerve regeneration using animals provides information to translational research and future therapeutic options available for humans [13]. Human and veterinary medicine can mutually benefit from research that applies a one Health perspective [14]. The enforcement of transdisciplinary strategies can improve the knowledge about the well-being of animals, humans, and the environment [15]. In PNI, the use of animals as models helps to achieve more results concerning different treatments, because clinical data from humans can have economic, practical, or ethical limitations. For that reason, the correct use of animals has proven valuable for later human clinical trials in PNI, which happens both in humans and animals. For that purpose, it is important to correctly choose the best animal model to use in each study to draw meaningful conclusions from animal experiments and translate them to human medicine. The One Health perspective in PNI benefits both human and veterinary medicine in the urge to obtain a treatment that enhances the outcome from PNI, especially in severe lesions.

The aim of this review is to address the PNI and focus on the treatments and advances achieved in peripheral nerve regeneration, especially using mesenchymal stem cells and biomaterials. Besides, the authors want to discuss the use of MSCs’ secretome as another possible therapy, as well as advantages and disadvantages from the conventional methods. Another approach is the One Health topic, since most of the studies use animals. Therefore, it is important to discuss the relevance of using animals to achieve an optimized treatment for both veterinary and human medicine. In addition, in vivo models provide more data about nerve regeneration using behavior and functional assessments. Since PNI can be chronic and lead to serious problems concerning quality of life, other recent studies will be mentioned where the diet and different types of exercise can help in reaching better outcomes.

## 2. Peripheral Nerve Anatomy

Each peripheral nerve is constituted by multiple longitudinal arrangements of axons, named ‘fascicles’, and covered by three layers of connective tissue. These layers support the fascicles and contain blood vessels that provide trophic support for the nerve fibers [1,5,7]. The epinerium is the outermost layer, which comprises all fascicles that form the peripheral nerve [7]. Areolar connective tissue, that allows the nerve contraction and expansion, is the main constituent of the epinerium [7]. All fascicles are coated by the inner portion of the epinerium, which contains blood vessels that irrigate the nerve and some adipose tissue [5]. The mechanical protection and anatomical shape are provided by the external layer that coats the entire nerve [5].

The middle layer is the perinerium, a thin and dense connective layer that envelops each fascicle individually. Therefore, helps to maintain homeostasis in the structure and to protect the endoneurial environment [5,7]. Lastly, the inner layer is the endonerium, a thin layer of collagen fibers that surround each axon within the fascicle [7]. It contains a thin network of micro vessels and capillaries and possesses great elasticity, but this layer assures little mechanical protection [5].

SCs are intimately associated with each individual myelinated axon [7]. These glial cells are capable of producing laminin-rich sheets of myelin, since they form a fatty multilayered membrane (several layers of SC membranes associated with secreted proteins) that isolate the axon. Myelin is very important to enhance the neural electrical impulses propagation rate along the nerve fiber. When the nerve fiber is demyelinated, the impulse moves in wave motions. On the other hand, in myelinated fibers the conduction occurs trough a saltatory propagation. Other myelin function that helps to assure a faster impulse is the increment of the electric resistance along the cell membrane. Besides the role in the myelination process, SCs are the main extrinsic mediators of peripheral nerve regeneration [7]. Another important cell type present in the peripheral nerve environment are the pericytes, which are contractile cells associated with the endothelial lining of microvasculature that modulate blood flow and capillary dilatation [7]. These cells help to maintain homeostasis both in the brain–nerve barrier and the endoneurial microenvironment [7]. The following figure (Figure 1) illustrates a schematic of peripheral nerve anatomy.

## 3. Type of Lesion and Scarring

The first classification system was introduced by Sir Herbert Seddon, in 1943, which described three classes of PNI: neuropraxia, axonotmesis, and neurotmesis. Class I, neuropraxia, is normally induced by ischemia or focal demyelination, due to traction or mild compression [7,16]. In this situation, sensory and motor connection is lost, and the conduction of nerve impulses is blocked without any anatomical damage to the *endoneurium*, *perineurium*, and *epinerium*. The recovery from this kind of injury is highly variable, but normally a full recovery is achieved in days or up to 3 months [17]. Class II or axonotmesis is caused by stretch, crush, or percussion and is more severe than neuropraxia. This is characterized as a disruptive lesion both of the axon and its myelin coating and the *perineurium* and *epineurium* integrity remains [5]. Following, within 24–36 h after PNI, Wallerian degeneration occurs distally to the lesion site [17]. In this type of injury, sensory and motor deficits occur, and the nerve conductions fail, distally to the injury site [17]. Class III or neurotmesis, is the most severe degree of PNI since it represents a nerve transection, where *endonerium*, *perineurium*, and *epineurium* are totally disrupted. This lesion has a poor prognosis, since it leads to the rupture of the axon, myelin sheath, and connective tissue [16,17]. In this case, recovery without surgery or other therapeutic option is almost non-existent [5].

In 1951, Sunderland extended Seddon’s classification from three to five degrees, based on increasing severity of damage in the nerve [5,7]. The first degree of PNI is the least severe and corresponds to neuropraxia from Seddon’s scale [2,17]. However, concerning axonotmesis, Sunderland divided this type of lesion in three different degrees: in the second-degree injury, the only structure that is disrupted is the axon, the endoneurial tubes, *perineurium*, and *epineurium* remain intact [2,17]. In the third degree lesion, the axon is disrupted and the endoneurium’s continuity is absent [17]. In the fourth degree, the *epineurium* is the only structure that maintains integrity, the axon, *endoneurium* and *perineurium* are injured. Finally, the fifth degree is the complete nerve defect as in neurotmesis, complete transaction of all layers [17].

The most recent update of PNI injuries was the introduction of a sixth-degree injury by Mackinnon and Dello, in 1988, (Figure 2) to the scheme of Sunderland. This last degree corresponds to the occurrence of mixed injuries in the same nerve [5].

### Events Following Peripheral Nerve Injury

After the lesion, axonal outgrowth is coordinated by many factors, including the infiltration of immune cells, SCs phenotype transformation and neurovascular regeneration [4]. Macrophages, neutrophils, endothelial cells, and SCs are very important in the different stages of the peripheral nerve repair mechanism. SCs proliferate, phagocyte debris in the lesion site and mobilize macrophages to promote the optimal regenerative surroundings [18,19]. Additionally, SCs form the myelin sheath along peripheral nerves and are responsible for the formation of the bands of Bunger [20].

The axons from the distal portion start to degenerate because of a mechanism known as Wallerian degeneration (WD), originally described by August Waller [21]. This degeneration begins with a flux of extracellular Ca^2+^ into the axon. For instance, the reduction of extracellular calcium or the ion Ca^2+^ channels blockage can attenuate the axon degeneration up to four days [19]. When increased, the intracellular calcium activates the calpains—proteases with the ability to cleave cytoskeletal proteins as the neurofilament. The discovery of a spontaneous mouse mutant, C57BL/*WLD*^S^, that exhibits very slow WD helped to understand this mechanism. WD is characterized by the breakdown and granular disintegration of the distal nerve stump after axon injury [19]. Upon an injury, the transport of nicotinamide mononucleotide adenyltransferase (NMNAT) by the cytoplasmic isoform, NMNAT2, from the cell body to axons to activate the nicotamide adenine dinucleotide (NAD+) is blocked. The short half-life of NMNAT2 explains the decrement of NAD+ levels that conducts to axonal disintegration [19]. WD results in the destruction of myelin and the enhance of SCs mitosis [22]. Following cytoskeleton and membrane degradation, the SCs surrounding the distal portion of the axon shed their myelinated lipids [22]. Then, the debris from this degradation are removed by macrophages and SCs. Following axonal fragmentation and debris removal, the axons have to regrow throughout a bridge formed between the proximal and distal stumps in the injury site. This bridge is mainly constituted of extracellular matrix components and inflammatory cells that help axons to regrow towards their original targets and its length depends on the injury size. When the axons reach the bridge’s edge, they initiate extending in the direction of the distal stump. Bands of Bunger are tubular structures present in the distal stump and important to guide the regrowing axons towards their original targets [20].

## 4. Advances in Peripheral Nerve Treatment

After a long regenerative period, full functional re-innervation without any complication is rare [2]. The process of regeneration in severe injuries is complicated and has limitations, which can result in the aggravation of muscular atrophy. For that reason, there is a need of surgery or other treatment, depending on the nerve gap and other factors [2]. The decision to perform a surgical intervention is complex: if the surgery is performed too soon, a lot of the potential for spontaneous recovery can be lost. However, confidence that the specific lesion has potential for a spontaneous recovery is not a risk-free decision since there is a period of time where the surgery can substantially improve recovery potential [5].

### 4.1. Surgical Treatment Methods

For PNI, the most common treatment method implies surgical approaches. For short gap PNI (<1 cm), neurorrhaphy is commonly used. This technique involves suturing the divided nerve, proximal and distal ends [1,2,10]. For that reason, it can only be used in short nerve gaps, since this method would cause excessive tension for a longer nerve gap, leading to bad regenerative results. In the 1970s, the use of fibrin glue to mitigate the effects of the suture emerged [1,2,10]. Comparing both techniques, fibrin glue provokes less granulomatous inflammation, decreases inflammatory response, has quick application, and the recovery is effectively the same. Nerve grafting is the most used technique for medium and large nerve gaps and recent studies showed that the reduction of stitches and the use of heterologous fibrin sealant can help minimize the trauma and, consequently, enhance nerve reconstruction [23,24].

Another strategy to reduce the number of stiches for reconstruction after PNI is the use of suture associated with heterologous fibrin sealant (HFS) [25]. HFS contains a fibrinogen-rich cryoprecipitate extracted from a buffalo’s blood (*Bubalus bubalis*) in addition to a thrombin-like enzyme purified from snake venom (*Crotalus durissus*) [25,26]. Furthermore, HFS is important to the healing process because the combination of fibrin with the proteins enhances angiogenesis, wound contraction, collagen synthesis and re-epithelization. The use of sutures can lead to inflammatory reactions such as granuloma and neuroma formation, so reducing the number of points used in the reconstruction is a way to minimize the damage and the inflammatory responses [24]. Leite et al. compared the use of HFS associated with a reduced number of stiches; one point, in comparison to the conventional three-point suture after ischiatic nerve injury in the rat model [24]. To prove the regeneration benefits of using HFS, the group used some regeneration evaluations such as Catwalk, electromyography, and morphometric analyses. The results showed that HSF sealant adjuvant to the suture had superior values concerning axons and nerve fibers diameter and area and better muscle weight. Therefore, these results suggest a protective effect of HSF and a decrease of trauma due to the stiches reduction [24].

#### 4.1.1. Autograft: The Current Gold Standard

To bridge large nerve gaps (>3 cm), critical nerve injuries and more proximal injuries, the most common strategy is the autograft, which consists in the use of autograft taken from the patient’s body from another nerve [2]. This technique implicates the use of functionally, but less crucial nerves, from various possible locations—such as, sural, superficial cutaneous nerve, or lateral femoral cutaneous nerve as a donor site [1,2,10,27]. To choose the most suitable nerve to harvest, the size of the nerve gap, as well as the location of the proposed nerve repair and the donor-site morbidity have to be considered [1,27].

The use of autologous nerve grafts functions as an immunogenically inert scaffold, which contributes to a stimulating and permissive environment, adhesion molecules, and neurotrophic factors, that promote nerve regeneration [2]. This strategy has some limitations such as donor site morbidity, need for an additional surgical procedure, loss of function from the donor site, scarring, possibility of painful neuroma, and the narrow viability of the graft tissue [1,2,10,27]. In addition, this technique has other disadvantages such as limited supply, the need for a second incision to harvest the graft tissue and fascicle mismatch [1]. A recent study tested the autologous nerve transplantation with an aligned chitosan fiber hydrogel (ALG), filled with a bioactive peptide, RGI/KLT. RGI peptide is derived from BDNF and is important in motor neuron outgrowth. KLT acts as vascular endothelial growth factor (VEGF), important in the angiogenesis process [28]. The purpose was to improve a 15 mm sciatic nerve defect and the complex compound had positive functional recovery results. SCs were inoculated into different groups and after 48 h, the SCs present in the ALG—RGI/KLT were orientated, in contrast with the cells from the control group that were disordered. The preliminary results prove that ALG helps to regulate the SCs directional growth. In addition, 12 weeks after the operation, the regenerated nerves were removed and stained. The number of regenerated nerve fibers was higher in the ALG–RGI/KLT group with greater diameter and thickness [23,28]. Besides this work, other alternatives and more complex formulations are being studied for better regeneration of the peripheral nerve.

#### 4.1.2. Nerve Allograft

When the nerve gap is above the critical size—which is about 3 cm in humans—and the graft length exceeds the accessible nerve autograft, other sources need to be taken into consideration. The use of a cadaveric or donor nerve, as an allograft, may be used as an alternative clinical option [2,27]. Nerve allograft contributes with guidance and viable SCs that facilitate axons connection, beneficial to reinnervate the target tissue. Nonetheless, this technique has some disadvantages: it requires a period of 18–24 months (approximately) of immunosuppressive therapy, post implantation, in order to prevent graft rejection and to allow the host axons and SCs to regenerate along the allograft scaffold [1,2,27]. However, due to this long-term immunosuppression, the patient is more susceptible to other problems, such as opportunistic infections or other systemic effects. Therefore, this technique is only used in the most severe injuries [2,27].

Allograft is easier to use, as it lacks donor site morbidity and presents unlimited supply and ready accessibility, compared to the autograft [1]. Nevertheless, there are still some problems regarding nerve allograft, leading to the emergence of artificial NGCs to enhance nerve regeneration and eliminate the necessity for immunosuppressants [2]. Given the disadvantages of the technique, new approaches are being developed to achieve better outcomes in PNI.

### 4.2. Emerging Approaches for Axonal Fusion

Photochemical tissue bonding is an emerging technique. It consists in cutting, on either side of a coaptation, the *epineurium*, followed by staining with a photoactive dye and using laser irradiation, with the purpose of affecting a watertight seal across the gap. The use of this method in a rat model with a sciatic nerve transection showed that the axonal fiber diameter and myelin thickness improved, in comparison with the traditional suture coaptation [29,30].

Another recent fusion technique, to restore nerve continuity and function, is the use of polyethylene glycol (PEG) to fuse the ends of transected axons. The PEG fusion has to be performed shortly after the injury to have positive effects. It consists in washing both cut ends of the nerve, with calcium-free hypotonic saline, and treating with methylene blue, an antioxidant, at least PEG is applied [29]. PEG works as a fusogen, with the proper alignment of the cut proximal and distal fascicles. In animal models, this technique demonstrated better results using PEG fusion over epineural suture repair, concerning the motor testing, histology, and other methods to quantify recuperation [29,31].

### 4.3. Non-Surgical Treatment Methods

Usually, non-surgical and surgical methods are combined, but the success rate in PNI injuries and in long nerve gaps (>3 cm) is still unclear [2,32].

The beneficial role of exercise on the nervous system is well stablished, promoting axonal growth and phenotypic changes in peripheral nerve architecture [13]. There is an increase of growth factors—stimulation of peripheral nerve regeneration, improvement of synaptic elimination, and release of neurotrophic factors that help to relieve neuropathic pain by stimulating the endogenous opioids release [5]. Aerobic exercise, such as swimming and walking, are the best options to help treat PNI. The volume of training, intensity, and the type of nerve injury are parameters that influence the effects of the exercise [13]. The combination of physical exercise with other therapeutic methods, such as electrical stimulation, seems to help achieve better outcomes in the regeneration of PNI [5]. In rats with peroneal nerve injuries, following 10 weeks using the treadmill for running at a 10 m/min velocity for 1.5 h, two times a day, the fast nerve fibers showed an enhancement in conduction velocity [13,33]. In addition, this exercise also reduced the levels of myelin-associated glycoprotein, which is an axonal growth inhibitor, thereby proposing an explanation for the axonal growth stimulated by the exercise [13].

Electrical stimulation (ES) is frequently used as a treatment, since low-frequency electrical stimulation has a positive effect on the nerve regeneration [1,5,34]. The combination of ES and physical activity, such as treadmill, also has positive effects [34]. In this technique, it is important that the frequency range is correctly selected, because higher frequencies can deteriorate and exacerbate atrophic muscle events [1,5]. In addition, it is essential to standardize some factors such as the duration of the electrical stimulation, consider the side effects on the healthy surrounding tissues and the variations provoked by the type of injury and extent of damage [1,34]. To accelerate nerve regeneration, ES can be used in combination with steroids or combined with surgical techniques [1,5]. One study where weekly electric stimulation was applied, using an implanted wireless device to stimulate rat sciatic nerve regeneration, showed acceleration in the recovery [29,35]. Thus, the ability to administer ES under local or regional anesthesia and using wireless implanted devices would be a promising technological advance and a new approach to common treatments [29].

Another non-surgical method is the magnetic stimulation, activating peripheral nerve regeneration by raising the number and diameter of the regenerated axons to promote functional recovery [5]. This technique is not totally understood but its effect is probably due to the NGF activity stimulation and diminished cytokine activity. The positive effects of this therapeutic method depend on the amplitude (0.3 to 300 mT), time (10 min/day to 24 h/day), and frequency of exposure (2 Hz to 2000 Hz). Additionally, in combination with mesenchymal stem cells, it promotes a neuron-like cells differentiation, interfering with the cell cycle [5].

Low-intensity ultrasound (LIU) can induce a positive response in peripheral nerve regeneration, probably through mechanical and thermal effects. The ultrasound promotes membrane diffusion movements and stimulates blood circulation, the release of brain derived neurotrophic factor (BDNF) and cell’s metabolism, as well as stimulating tissue nutrition [5].

Phototherapy, a low power radiation, also promotes axonal regeneration. This technique has positive results, such as diminished scar tissue formation, reduced degenerative process and stimulation of myelination and axonal regrowth [5].

Photobiomodulation therapy (PBMT), using low-level infrared light spectrum lasers, is also considered a therapeutic advance. Laser therapy is considered a non-invasive treatment that activates a photochemical reaction in the cells, resulting in the increase of DNA and RNA synthesis. Consequently, protein synthesis occurs alongside cell proliferation, promoting changes in nerve cell action potentials [34]. The effects are associated with tissue biostimulation. Both in vivo and in vitro studies have shown that this technique enhances the functional recovery and regeneration of the injured peripheral nerve by stimulating SC proliferation and expanding the axonal diameter [34]. Nevertheless, there are still some issues related to the application parameters, since there is no standardization in treatment with PBMT [34,36].

#### 4.3.1. Growth Factors

Nerve growth factors (GFs), are molecules released naturally in the injury process and directly enhance nerve regeneration. For that reason, mimicking their release can be an important tool for nerve survival, differentiation, and growth [5,27,37]. Therefore, adding these factors is another therapeutic approach that can improve the microenvironment of guide tubes, making them more permissive for axonal regrowth [27]. On the other hand, the artificial application of GFs as a therapeutic option is complex, because of their vast biological activity, making it crucial to be administered in small and precise doses. Other limitation is the short biological half-life and the pleiotropic effects. To improve this technique, in addition to a better characterization of the molecule’s profile, delivery systems are being investigated [27,37].

There are diverse GFs and each one has different actions, such as promoting functional regeneration, supporting SC migration and axonal elongation. There are three different groups of molecules with these effects: the neurotrophins (nerve growth factor (NGF), BDNF, neurotrophin-3, and neurotrophin-4), neuropoietic cytokines and the glial-cell-line-derived neurotrophic glial-cell derived neurotrophic factor (GDNF), ciliary neurotrophic factor) [5,27,37]. From these, NGF is the most explored so far, since it is commonly present in healthy nerves, its expression raises in injured nerves, and has essential functions to enhance neuron growth and survival [5]. GFs biological activity depends on their interaction with some target receptors that are expressed on the neuronal cells surface [27]. The main reason to use these factors is to enhance axonal elongation and SC migration by providing a neurotrophic and neuroprotective environment after nerve injury [27]. In addition, GFs can reduce the p38MAPK expression, a pathway activated in early stages of lesion in the PNS and associated with cell death [5].

GFs can be used in combination with NGCs, directly into the conduit lumen or through different carriers [27]. The selection of neurotrophic factors and the suitable delivery method for these molecules are important to achieve good regenerative outcomes. The delivery of GFs from the conduit wall can be achieved with methods such as conjugation, adsorption, and physical entrapment in the wall. To achieve a beneficial effect on nerve regeneration it is important to consider the optimal dose and GF release kinetics, for standardization and NGCs optimization. Support cells are attracted by the growth factor delivery into the conduit lumen that increases axonal regeneration and contributes to a better microenvironment for repair of nerve gaps [27]. Cristina et al. tested two methods (crosslink and adsorption) and compared their ability for controlled delivery and retention of two different GFs, GDNF, and NGF. To assess the outcome of the NGCs delivering the GFs, this study used a 100 mm sciatic nerve injury model in rats. The results proved that the controlled release of GDNF and NGF from NGCs enhanced nerve repair [38]. Other recent study hypothesized that mineral coated microparticles (MCMs) could generate a sustained release of NGF and GDNF inducing nerve regeneration and a significant improvement in functional recovery. This study used a rat model of the sciatic nerve and the in vivo results in the rats with nerve grafts incorporated with MCMs. Releasing both the GFs showed significantly more myelinated axons, compared with the nerve grafts without growth factor treatment [39].

BDNF is a neutrophin that regulates the synaptic function and has potential to enhance nerve regeneration and preservation. In addition, this molecule is important in the reduction of food intake and, consequently, body weight, and anorexigenic regulation. For that reason, reducing energy intake has been proved to enhance the neuronal process and the synaptic activity. It has been shown that, in rats with energy limitation, the peripheral nervous structures were markedly preserved, and the degenerative events diminished [40]. Since it is improbable to advise patients to limit their energy uptake and recommend a low-calorie diet, on peripheral nerve injuries, this result does not show a promising therapeutic option.

#### 4.3.2. Mesenchymal Stem Cells

Following PNI, axonal outgrowth depends on many factors, such as the infiltration of immune cells, neurovascular regeneration, and the phenotype of SCs transformation [4].

SC-based therapies have been employed in many studies to enhance nerve regeneration, providing the nutrition and support for axonal growth [4]. Furthermore, over the years, various studies have focused on methods to promote axonal regeneration, without the consequence of sacrificing other healthy functioning nerves, as in the technique of using autologous nerve graft. The need to improve these technique limitations brought the urge to study synthetic nerve conduits, that will be discussed further on, and mesenchymal stem cell-based therapy, which is thought to be a promising strategy to PNI and a number of other diseases [4,16,18,41,42]. MSCs can promote regeneration by signaling through cell-to-cell contact, cell differentiation into tissue-specific cell types and release of neurotrophic factors [18].

MSCs are multipotent stem cells that can be harvested from numerous sites of the body, such as adipose tissue, dental pulp, bone marrow, umbilical cord, and amniotic fluid [16,41,43,44]. The different tissue origins where MSCs can be harvested explain different characteristics among these cells, such as distinct cytokine expression profiles. Besides, some origins are better than others, to a certain clinical application [41]. In addition, MSCs are undifferentiated, easily isolated, plastic adherent when maintained in standard conditions, and have low immunogenicity. For that reason, MSCs may be transplanted allogenically with minimum consequences [42,43,44]. Furthermore, MSCs can differentiate into SCs, one of the reasons why these cells are so valuable to PNI [41,44]. Mesenchymal and Tissue Stem Cell Committee of the International Society for Cellular Therapy human MSCs characteristics are determined by expressing cluster of differentiation CD105, CD73, and CD90 and lack of expression of CD45, CD34, CD14 or CD11b, CD79 or CD19, and HLA-DR surface molecules. These cells must differentiate in vitro, in the presence of adequate differentiation media, into osteoblasts, adipocytes, and chondroblasts [42]. Progressively, MSCs are considered to act through paracrine effects by the release of extracellular vesicles, which contain components like cytokines and miRNA and can enhance axonal regeneration [41].

In peripheral nerve regeneration, MSCs have an important role due to their ability to produce and release neurotrophic factors that induce axonal growth, their self-renewal capacity, the ability to differentiate into myelinating cell lines and into Schwann-like cells [16,43,44]. MSCs have been studied as an important tool for tissue repair because of their capacity to migrate to the site of injury and secrete bioactive factors. This contributes to the paracrine activity, and the ability to suppress the inflammatory response injury, which promote nerve regeneration [44]. Additionally, MSCs prevent cell death by diminishing the expression of pro-apoptotic factors while increasing anti-apoptotic activities [44]. Some of the multifunctional characteristics of MSCs make them ideal for therapeutic application. For these reasons, in this review, different types of stem cells will be addressed, summarizing the literature supporting MSCs use in PNI.

##### Bone Marrow Mesenchymal Stem Cells

Bone marrow mesenchymal stem cells (BM-MSCs) are one of the most studied cell types in this field [18]. These multipotent cells can differentiate into mesenchymal lineages (fat, bone, muscle) but also into non-mesodermal lineages—such as astrocytes, neurons, and Schwann-like cells—under certain environmental conditions [18,43,45,46]. BM-MSCs produce and secrete neurotrophic factors (NGF, GDNF, ciliary neurotrophic factor and BDNF) and extracellular matrix components (collagen, laminin) that enhance regeneration and angiogenesis [18,43,46]. In comparison with other MSCs sources, these cells also have some limitations such as the invasive and painful extraction of bone marrow, some ethical controversies and decreased capacity for proliferation and differentiation [43,46].

Some studies, particularly using the rodent model, found that BM-MSCs application in combination with conduits and nerve grafts showed increased axonal regeneration, improved wet muscle weights and walking track scores [18,43,47,48]. However, BM-MSCs have also been studied in larger animal models, such as rabbits and sheep [18]. One recent study tested BM-MSCs, platelet rich plasma with a biodegradable scaffold in nerve gaps with 1 cm on the sheep radial nerve. Results proved that, 6 months later, morphometric and neural conduction improved, showing myelinated nerve fibers both in the proximal and distal segments of the nerve when treated with BM-MSCs combined with a scaffold [49].

##### Adipose-Derived Mesenchymal Stem Cells

Adipose-derived mesenchymal stem cells’ (Ad-MSCs) source is the stroma of fat tissue, making them accessible and abundant [45,50]. In comparison with BM-MSCs, Ad-MSCs can be collected in a liposuction, a less invasive manner and in higher quantities [17,50,51]. The capability of differentiation and proliferation are much higher than other sources of MSCs [17,51]. In addition, these cells have low immunogenicity and are able to maintain long-term plasticity and phenotype in vitro [51]. Another important advantage is that location and age of the donor does not influence the therapeutic effect. For these reasons, Ad-MSCs are attractive for regenerative medicine, tissue engineering, and peripheral nerve regeneration [50,51].

These cells follow the criteria of the Mesenchymal and Tissue Stem Cell Committee and can secrete various cytokines, growth factors (VEGF, granulocyte colony-stimulating factor, hepatocyte growth factor) increase myelin sheaths formation, suppress inflammation, guide outgrowing axons, and differentiate into a SC phenotype, which indicate multiple advantages for regenerative approaches [17,45,50,51]. The transplantation of canine Ad-MSCs, after crush lesion of the rat sciatic nerve, accelerated functional motor recovery, proved by the results obtained from electromyography and sciatic function index gait analysis in some studies [17,45,50].

##### Dental Pulp Stem Cells

Dental pulp stem cells (DPSCs) are a suitable stem cell source because these cells can be obtained from orthodontically extracted premolar and impacted third molars, which does not raise ethical controversy nor extra harm [46,52,53]. These MSCs have ectodermal origin and migrated from the neural tube region into the oral region and differentiated [45,52,54]. DPSCs are multipotential cells, and when exposed to different differentiation media these cells proved their osteogenic, neurogenic, adipogenic, chondrogenic, dentinogenic, and myogenic differentiation capacity; as well as plasticity, self-renewal, and high proliferation rate [45,52,53].

These cells express several markers including the mesenchymal (CD13, CD29, CD44, and CD146), specific neuronal markers and BM-MSCs markers [53,54]. DPSCs ability to differentiate into Schwann-like cells, the capacity to promote myelin reconstitution and axonal regeneration are some of the reasons why these cells can be used in PNIs [46,52]. In addition, these cells can secrete neurotrophic factors that provide trophic support for a better axonal regeneration and neuroprotection [45,52]. After a crush in the sciatic nerve of rats, Okuwa et al. transplanted DPSs at the lesion site. Fourteen days after the surgery, the recovery of motor function was shown through a higher ratio of tibialis muscle weight. The morphological and functional findings proved a good correlation between the use of these cells and peripheral nerve regeneration [55].

##### Fetal Tissue-Derived Mesenchymal Stem Cell

Amniotic fluid and umbilical cord (both considered as fetal tissues) are the most primitive sources of MSCs [46].

The umbilical cord is a relevant source of MSCs, both from umbilical cord blood and the Wharton jelly [18,46,56]. Therefore, umbilical cord derived mesenchymal cells (UC-MSCs) can be harvested in a non-invasive way from postnatal tissue and possess high capacity to expand ex vivo [41,46,56]. These cells have previously demonstrated the potential to express pluripotent stem cell markers, such as Nanog, Sox 2, ABCG2, Oct4, and the neuroectodermal marker nestin [18,56]. Additionally, UC-MSCs are multipotent stem cells with self-renewal ability and the capacity to differentiate into osteocytes, chondrocytes, adipocytes, and neuronal cells [56].

In comparison with other sources of MSCs, these cells have decreased expression of HLA-I and possess greater paracrine effect [18,41,57]. In addition, through sequential treatment with B-mecaptoethanol and some cytokines, UC-MSCs can adopt a Schwann-like phenotype. These characteristics make them able to potentiate peripheral nerve and axonal regeneration [41]. Other potential advantages are the low probability of resulting in rejection, the off-the shelf availability, and the lack of related ethical considerations [18,56].

UC-MCSs have been investigated in various studies including in sciatic nerve injury, optic nerve injury and recurrent laryngeal injury, proving to improve recovery and to promote neurogenesis [18,41,56].

Wharton’s jelly is an easily accessible, noncontroversial source of MSCs and has unique properties. Additionally, amniotic mesenchymal stem cells (AMSCs) are relatively low immunogenic cells that derive from the avascular amniotic mesoderm [46,57]. The collection of amniotic cells can easily be accomplished by prenatal testing and the amniotic membrane can be collected using standard isolation methods after cesarean section [57]. These cell administrations improve functional recovery after nerve injury for different reasons. For instance, AMSCs secrete angiogenic factors, such as VEGF, express chemokine genes and receptors (CCR3, CCR2, CCR5), increase cell migration, endothelial trans differentiation properties and engraftment [46]. Additionally, to the MSCs characteristics, these cells have the capacity to differentiate into neural tissue under specific conditions [46]. Besides, these cells have some advantages, in comparison with other stem cell sources, such as low/no-tumorigenicity, multi-differentiation and renewal potential, no ethical or legal concerns, and impressive paracrine effects [58].

##### Skeletal-Muscle Derived Stem Cells

Skeletal muscle-derived stem cells (Sk-SCs) can be obtained from skeletal muscle satellite cells and have the capacity to differentiate into various lineages such as adipogenic, osteoblastic, neuronal, myogenic, and glial [18,46]. These cells represent an opportunity in peripheral nerve regeneration. In a study of long nerve gap injury in a murine model, Sk-SCs differentiated into Schwan cells and improved the *endoneurial* and *perineurial* architecture, which suggests the ability to reconstruct the muscle-nerve-blood vessel unit [18,46].

Muscle progenitor cells (MPC), have well-characterized markers and are an easily accessible type of cells, simple to manipulate in culture. Reut et al., evaluated the use of MPC overexpressing the NTF genes in a mouse model of sciatic injury, and proved that the combination of these cells with such important trophic factors enhance the peripheral nerve regeneration [23,59].

##### Hair Follicle-Associated Pluripotent Cells

Hair follicle-associated pluripotent (HAP) cells are in the hair follicle, express nestin, and are being studied because of their involvement in the formation of the hair follicle sensory nerves [18]. These cells can differentiate into glial, neuronal lineage, smooth muscle myocytes, melanocytes, keratinocytes and are considered pluripotent. The problem concerning the use of HAP is that they remain pluripotent in regenerative models, with neuroglial differentiation making it difficult to assure a specific differentiation into one of the possible lineages [18].

##### Neural Crest Stem Cells

Neural crest stem cells (NCCs) have an important role during neurogenesis, especially in the development of the spinal cord and brain, which occurs during embryogenesis [43]. In the human adult, these cells are located in the subventricular regions and in the hippocampus. These cells maintain their multipotency and can differentiate into astrocytes, neurons, and oligodendrocytes [18,45]. Additionally, these cells have already been identified in both postnatal adult and embryonic tissues, which include the bone marrow, cornea gut, heart, sciatic nerve, and also the skin [18]. Transplanted neural stem cells can differentiate into neurons and Schwann-like in injured peripheral nerves, secrete BDNF, fibroblast growth factor, insulin-like growth factor and NGF, restore angiogenesis, form myelin, and enhance nerve growth [17].

For those reasons, some studies suggest that the application of NCCs in the injury site, both after acute and chronic PNI, can enhance nerve regeneration. After sciatic nerve injury, one study used diverse methods to quantify the result of applying NCCs transplantation in the lesion site. These cells showed a great potential in regeneration potency, especially at the fifth passage [23]. However, the limitations of these cells include the technical difficulties associated with cell harvest, tumorigenic risks and the urge to differentiate directly into specialized neural cell lines [18,43,45].

##### Skin-Derived Precursor Cells

Skin-derived precursor cells (SKPs) originate in dermal papilla and have the ability to differentiate into neurons, glia, and smooth muscle cells. These cells have already proved to be beneficial in sciatic nerve repair and in the ability to improve regeneration after chronic denervation [18].

##### Olfactory Mucosa Mesenchymal Stem Cell

Olfactory mucosa mesenchymal stem cells (OM-MSCS) are collected from the olfactory mucosa, which anatomical location is ideal for both *postmortem* and *antemortem* collection in various species of clinical interest. These cells have fibroblast-like morphology, clear plastic adhesion, and express specific surface markers with no expression of hematopoietic markers, as expected from MSCs. In addition, OM-MSCs can generate new fibroblast colonies from single cells and are able to differentiate towards adipogenic, chondrogenic, osteogenic, and neurogenic cell lines [22,60].

A recent study used chitosan NGCs and OM-MSCs in different therapeutic approaches to enhance rat sciatic nerve injury, following neurotmesis. All the therapeutic groups receiving chitosan NGCs and the cells had best final results in the functional assessment, kinematic analysis and histomorphometric evaluations, 20 weeks after the inflicted injury. These results demonstrate promising effects of the use of OM-MSCs in PNI [61].

#### 4.3.3. Secretome

All the previous mentioned MSCs can secrete bio-active factors, defined as secretome that comprises chemokines, cell adhesion molecules, cytokines, lipid mediators, growth factors, hormones, micro vesicles, and so on. The constituents are considered the main participants in tissue repair and regeneration, through their paracrine capacity that mediates cell-to-cell signaling [44,51,62]. The secretome involves two components: a vesicular fraction composed by different types of vesicles, crucial in the delivery of microRNAs and proteins, and a soluble fraction constituted by cytokines and chemokines [62]. For those reasons, recent studies reinforce the effectiveness of MSCs secretome therapeutic potential in different diseases [44,51].

This new line of research contributes some advantages over cell-based applications. Secretome has lower immunogenicity compared to living and proliferating cells, and the analysis of secretome safety, dosage, and potency is simpler [51,62]. The transcriptome and proteome are very different in MSCs populations, so the secretome profile is also heterogeneous [44]. The use of various types of MSCs and methods to develop secretome, for the treatment of various diseases—including PNI—are receiving more attention in recent time [63]. Amir et al. incorporated on a 3D-polycaprolactone scaffold the BM-MSCs’ secretome to examine the efficiency in an axotomy in the rat model. In this study, there were four groups, one control, the second with just the axotomy (10 mm of the nerve was removed), the third with axotomy and only the scaffold and the last with the scaffold incorporated with the BM-MSCs’ secretome. After 12 weeks of treatment, the results showed that the sciatic nerve was significantly restored, especially in the third and fourth groups. In addition to the advantages from using secretome, in this group the levels of NGF and BDNF were increased [64].

Another recent study used Schwann-like cells derived from differentiated human embryonic stem cell-derived neural stem cells secretome to explore the influence in angiogenesis and nerve regeneration. After sciatic nerve crush injury, using the rat model, this secretome was used to study its role in nerve regeneration. In comparison with the control group, the animals treated with the secretome showed better gait recovery and the gastrocnemius wet weight was higher, after 14 days [65]. Martins et al. investigated the UC’s secretome on axonal elongation of CNS neurons. The results proved that this secretome promotes axonal outgrowth in primary cultures of rat embryonic hippocampal and cortical neurons [66]

#### 4.3.4. Gene Therapies

In recent years, it was noticed that some alterations in the genes (insertion, alteration, or removal), in some cells, can be used to treat diseases [5].

The modulation of regulatory proteins and neurothrophic factors can enhance post-surgical results. The main goal of gene therapy in PNI is to increase axon regeneration and the most efficient technique for nerve regeneration is gene delivery [5]. For instance, injections of a viral vector (every 5–8 mm) has been proved to be highly successful in the gene transduction of the rat’s sciatic nerve [67]. In addition, this technique can be efficiently performed at the site of repair. Afterwards, retrograde transport of the viral vectors distributes it to the cell body, a place where gene transcription may be changed [29]. Animal models were successfully transfected with genes encoding neurotrophic factors, and the recovery was achieved more rapidly. Besides, SC is another target for gene therapy that has shown improvement in regeneration. The use of ciliary neurotrophic factor, a factor unique to the SC, in a mouse model of sciatic nerve crush injury, was transduced using an adenoviral-associated vector that demonstrated an enhancement in axonal regeneration [29,68]. Another hypothesis concerning its use is the overexpression of c-Jun, a transcription master switch which is upregulated in case of an injury [29]. In short, gene therapy is a promising method for the customizable delivery of neurotrophic factors, but there is still a long way to go for this technique to be fully effective and safe [5,29].

#### 4.3.5. Pharmacological

To avoid scar formation in peripheral regeneration barriers can be surgically created between the surrounding tissue and the injured nerve, complemented by pharmacological agents as steroid hormones [5]. They can promote myelination and have neuroprotective proprieties [34].

Corticosteroids are also a possible treatment option for nerve injury. The study of Bernstein et al. demonstrated that the administration of oral corticosteroids enhances the motor and sensory recovery after iatrogenic PNI [34,69]. These substances act by inhibiting fibroblast growth and the migration/phagocytic action of granulocytes, selectively [5]. For example, dexamethasone, frequently used to reduce edema and the neural inflammation effects, turns up to enhance nerve recovery after injury [5]. The animals treated with these substances are associated with higher recovery rates [5,34]. Thus, the use of hormonal therapies is not commonly used in the treatment of PNI because these substances can have side effects.

Erythropoietin (EPO) is an endogenous hormone, a hematopoietic agent, especially used in the treatment of anemia. EPO has shown neuroprotective properties both in CNS and PNS. In addition, EPO and its receptor are present in a variety of non-erythroid cells, and for that reason have an impact in many biological functions throughout the body [34]. For instance, EPO is secreted and produced by diverse parts, like neurons of hippocampus, internal capsule, cortex, and midbrain. Besides, the receptor is also expressed in the radicular nerves in the PNS on the myelin sheath. For those reasons, EPO is demonstrated to have a positive effect in the sciatic function index of rats and mice, after crush injury [34,70]. Besides, the effects were observed in different times of use in the treatment. The mechanism of action is not well known but it is thought that EPO promotes the expression of myelin genes, such as MPZ and PMP22, maintaining more myelinated axons at the site of injury [34].

4-aminopyridine (4-AP) is used in the symptomatic treatment of multiple sclerosis. It is a broad-spectrum potassium channel blocker, capable of improving neuromuscular function in patients with other demyelinating disorders [5]. The mechanism of action results in the calcium influx that stimulates the release of the neurotransmitter thus causing synaptic transmission and, ultimately, direct effects on the muscle [34]. The capacity of 4-AP increasing cell-membrane excitability and, consequently, impulse condition shows that, relevant doses of this molecule shortly after injury, enhance global function recovery, improve nerve velocity of conduction, allowed a better and faster behavioral and motor recovery and promotes remyelination, in established mouse model after PNI [5,34,71]. Besides, 4-AP appears to be axonoprotective and myeloprotective, which can be helpful and contribute therapeutically to recovery due to excitatory molecules that can stabilize impulse conduction [34]. Nevertheless, there are some side effects, and more studies are needed to guarantee safer use of this substance.

#### 4.3.6. Nutritional Therapy

Some nutrients can have a role in preserving nerve health and function and enhancing the recovery of an injured nerve. Nevertheless, there are several studies about the mechanisms by which both nutritional factors and nutrients have an impact [40].

The nervous system is extremely rich in lipids that are fundamental for several functions. Poly-unsaturated acids (PUFA) are recommended to be balanced in each individual diet. For example, a study in Wistar rats, proved that the intake of n-3 PUFAs has positive effects on the regeneration of a sciatic nerve injury [40,72].

It has been proved that vitamins and minerals have a fundamental role in the prevention of oxidative stress, reducing neuroinflammation and providing protection against cellular injury [5]. In addition, these molecules may have helped in the production of endogenous neurotrophic factors, like BDNF, that will enhance nerve repair [40].

Vitamin B_6_ supplementation mitigates symptoms of neuropathy and numbness. Vitamin B_12_ enhances the number of SCs and promotes axonal regeneration by improving the remyelination [5,40]. Furthermore, there are studies proving that high doses of this vitamin have potential to improve the PNI treatment, in a rat sciatic nerve injury model and that this vitamin can upregulate genes related to various GFs [5,73]. Another promising result in rats was the local implantation of methylcobalamin in nanofiber layers. The results demonstrated a positive synergistic effect, where both motor and sensory functions improved, with an enhancement of nerve conduction velocity and myelination, after sciatic nerve injury [40,74].

Ascorbic acid (Vitamin C) helps SC myelination of peripheral axons, both in vivo and in vitro, since co-cultures fail to produce myelinated segments without vitamin C [75]. Tyler et al., using a mouse model, showed that dietary vitamin C deficiency impaired, in early stages, the peripheral myelination [75]. Besides, this study investigated that the intake of this vitamin was fundamental for the expression of periaxin and myelin basic protein, both important components of the myelin sheath. For those reasons, vitamin C can promote SC myelination, not only by collagen stability, but also through direct epigenetic regulation, making this substance critical throughout development and during remyelination, after PNI [75].

Folic acid and even minerals play a role in cognitive functions and enhance neurogenesis. Moreover, phenolic components, such as curcumin, are potent neuroprotectants. Curcumin effect in various dosages on sciatic nerve regeneration has been investigated, demonstrating that a higher dose has better results in the recovery of nerve function and enhanced the remyelination [5,40]. Other substances, such as resveratrol, *Alpinia Oxyphylla* (important herb), epigallocatechin-3-gallate (catechin of green tea), therapeutic effects on PNI are being studied and showing positive effects [40].

Medicinal plants, such as *Achyranthes bidentata*, *Astragalus membranaceus*, *Panax ginseng*, *Hericium erinaceus*, *Acorus calamus*, *Curcuma longa*, and *Ginkgo biloba* have proven ameliorating effects in neuropathic pain [76].

Another important factor for neuroprotection is the early application of antioxidant therapies. For instance, the use of alpha-lipoic acid demonstrated positive effects, as this thioctic acid stimulates an increase in endogenous antioxidants (SOD and CAT). Therefore, alpha lipoic acid can be considered in the treatment of PNIs [5,40].

### 4.4. Nerve Guide Conduits

Other therapeutic option is the use of NGCs. NGCs are tissue engineered tubular structures that serve as bridge between the proximal and distal ends of the injured nerve [1,2,10,45]. These can be made of natural and/or synthetic biopolymers, which are planned to possess the necessary biomechanical and mechanical cues for neural regeneration [1]. Most of the disadvantages described regarding the use of autografts can be avoided by the use of NGCs [1,9]. NGCs allow trophic and structural support for both the nerve ends, supporting the regrowth of axons alongside the conduit and the invasion of the surrounding tissues. Additionally, these conduits can have neurotrophic factors, proteins, and even extracellular matrices, throughout their inner surface, that can increase the axon regeneration [1,2,45]. Besides, the most valuable improvement of a conduit is its capability to provide a better microenvironment for neuronal recovery. For peripheral nerve repair, there is a need for compliance between support cells, conduits, growth factors, and physical stimuli [45]. Therefore, NGCs act as a guide for nerve regeneration, since they allow cells to adhere and migrate, thus achieving the desirable regeneration [45].

An ideal NGC has some requirements, including structural features to longitudinally align the regeneration axons, biomimetic architecture, mechanical properties to provide structural support, adequate permeability for trophic support, wall thickness, specific conduit diameter, compliance, neuro inductivity, low toxicity, electrical conductivity, biodegradability, biocompatibility, and flexibility [1,2,45,77]. In 1982, the first application of NGCs was made, using non-resorbable silicon tubes to repair a 6 mm nerve gap. Since then, researchers have been trying to obtain the ideal NGC, improving the design, fabrication process and materials, especially for complex defects [1,2]. Different artificial nerve conduits have already been studied and are commercially available and approved by the Food and Drug Administration and European Medicine Agency, especially for short nerve lesions [45,77].

After manufacturing, the biomaterial is tested in vitro models with the purpose to access its cytotoxicity, biocompatibility, genotoxicity, degradation, cell proliferation, and interaction between cells and the biomaterial [77]. The use of in vitro assays is crucial for the study, making it more easily standardized and reproducible and aids reducing the number of animals used in the in vivo testes taking into consideration the “3R” philosophy (reduce, replace, refine) [77].

Kornfeld et al. recently used spider silk as a potential material of choice to overcome the limitations of nerve gap repair of 3 cm. This material has been proved to be an effective guidance architecture for SCs and to allow nerve regeneration both in rodents and ovine models. One recent study examined the time course of axonal regeneration rate using spider silk nerve implants and compared to autologous nerve grafts in a 6.0 cm nerve defect in adult sheep [32]. The results obtained proved that the conduits were as effective as the autologous nerve implants, without the potential morbidity of autologous nerve grafts (sensory loss, neuroma formation) [32].

#### 4.4.1. Transplantation of Cell-Seeded Nerve Conduits—Peripheral Nerve Regeneration

The previously mentioned MSCs-based therapies can be used in association with NGCs, to promote nerve regeneration. The successful outcome of this therapy depends on the production of extracellular matrix proteins and trophic factors, to create a favorable environment for axonal outgrowth. In addition, the differentiation into SC lineage benefit myelination of regenerating axons [45]. Numerous types of cells, previously described, can be applied inside NGCs. It is important to choose the NGCs depending on the therapeutic option in study. For instance, the structure is fundamental for the cells to enter and proliferate when seeding MSCs in NGCs.

One recent study used canine Ad-MSCs in 3D-printed polycaprolactone NGCs in a sciatic rat model, 12 mm gap injury. In addition, a scaffold composed of heterologous fibrin biopolymer derived from snake venom was used to retain Ad-MSCs into the internal wall of the NGC. This material is non-cytotoxic, biodegradable and the sealant properties are perfect to avoid Ad-MSCs loses from the NGCs. The positive functional and electrophysiological locomotor recovery, after 8 and 12 weeks, proved that different combinations can be a desirable approach to overcome PNI [78].

Jahromi et al. developed a novel strategy for PNI with combination of gold nanoparticles and BDNF—encapsulated chitosan in lamini-coated nanofiber of poly (lactide-co-glycolide) (PLGA) with rat Ad-MSCs suspended in alginate. In this study, alginate was used as a hydrogel to retain the Ad-MSCs, providing an environment that prevents the leak out of the cells. This therapeutic technique was tested in a rat sciatic model and the results showed that this complex NGCs could enhance nerve regeneration [79].

Recently, a group developed a tissue-engineered NGC coated with induced pluripotent stem cells—derived neurospheres in a rat sciatic nerve model, 5 mm injury. This study had three different groups, one with the aforementioned therapeutic option, the control, and other with the autograft. The results from the autograft were slightly better than the therapeutic option. However, the group with the cells showed higher axon number and good outcomes making this other promising cell-based therapy using NGCs in the PNI treatment [80].

Bagher et al. loaded different concentrations of Hesperidin, that is a natural bioflavonoid, into cross-linked alginate/chitosan hydrogel and evaluated its characteristics. Following the cytocompatibility tests, the achieved hydrogel was administrated in a sciatic nerve injury in a rat. Results from weight loss, walking-foot-print analysis, hot plate latency teste, and other proved that the NGC loaded with hesperidin improved sciatic nerve regeneration [81]. Thus, recent studies show that NGCs combined with other substances can significantly improve nerve regeneration and are therefore promising in the treatment of PNI.

One study used the rat model after sciatic nerve neurorrhaphy and tested the combination of human embryonic stem cells, genetically modified to overexpress basic fibroblast growth factor, with HFS scaffold. The aim was to enhance regeneration and neuronal survival using autografting with end-to-end neurorrhaphy [82]. To determine if the use of this combination was a good solution for PNI, the authors used the walking track analysis, immunohistochemistry, and the von Frey test. The results proved that the application of this scaffold incorporated with human embryonic stem cells, on the site of injury, successfully regenerated sensory and motor fibers [82].

Other groups used a fibrin sealant (FS) derived from venom of the snake *Crotalus durissus terrificus*, as support for MSCs to enhance regeneration [83,84]. Spejo et al., applied a FS scaffold incorporated with BM-MSCs in filling the gap formed by the induced lesion at the spinal levels (L4, L5, and L6) in the rat model [83]. The results demonstrated that this combination caused greater expression of anti-inflammatory cytokine IL-1ß and proinflammatory M1 macrophages. However, the environment provided by the scaffold could have altered the BM-MSCs properties guiding to a proinflammatory milieu that resulted in the absence of motor improvement, in comparison with the group only using MSCs as a treatment [83].

#### 4.4.2. Advances and Next Generation Nerve Conduits

There are some conventional methods used for accomplishing hollow NGCs, including injection molding, melt extrusion, braiding, and crosslink. In all these techniques there is a limitation, the generation of simple macroscale conduit structures, with poor geometric accuracy. On the other hand, advanced techniques such as inkjet printing, 3D printing, fused deposition modeling, electrospinning, and three-dimensional bioplotting have been investigated to fabricate complex NGCs [27]. Electrospinning is a commonly used method to fabricate nano- or microfibers. It can mimic the natural extracellular matrix on a fiber structure, and optimize and manipulate mechanical, biological, and kinetic properties, in order to enhance cell–substrate interactions [27].

NGCs engineering achievements can help to guide the axons regeneration and incorporate supportive cells, genetic manipulators, GFs, and electro conduction [10]. The traditionally available conduits are simple hollow tubular constructs that do not possess the attributes of an autograft. Recently, major studies have been made directly to design scaffold characteristics to improve the nerve guides efficiency [27]. The different types of possible alterations implicate the addition of luminal fillers as guidance arrangement, neurotrophic factors, supportive cells, and conductive polymers inside the tube, with the purpose of improving the efficiency of regeneration in both small and large gaps [27].

Recently, technology involving 3D printing helps the fabrication of customized NGCs. 3D printing has advanced the field of regeneration by reproducing a 3D structure that mimics the natural extracellular matrix. However, there are some limitations such as low printing resolution, limited speed of printing and lack of a dynamic environment. For that reason, 4D printing is one of the most progressive methods of NGCs fabrication for the next generation of NGCs [27]. 4D printing technology can help to fabricate structures that can transform their shape, called the shape-morphing effect [85]. For that reason, these materials can transform and recover its shape depending on the external stimuli (light, temperature, and humidity) mimicking the physiological changes in the body [85].

To summarize this chapter about the available treatments for PNI, Figure 3 divides this section in two major options, surgical and non-surgical, showing the different possibilities in each one. 

## 5. In Vitro and Animal Models

### 5.1. In Vitro Models

Diverse distinct factors of the peripheral nerve regeneration mechanisms can be reproduced in vitro [86]. In the design of a study, it is important to begin with the in vitro models before trying to prove those concepts in in vivo models, mainly respecting the 3Rs. For these studies, cell line-based models can be used without any ethical impact.

Cell lines have clear ethical advantages since they replace any further animal use. In addition, these cells can be derived from human tissues, which enhances the translational research perspective towards the clinic [86]. In peripheral nerve research, cells cannot be used to anticipate the in vivo behavior of neurons and SCs because of the structural and anatomical complexity of the organs involved. The limited in vitro reproduction of the in vivo mechanism is the biggest limitation and the real reason for the need to use in vivo models.

Organotypic cultures are in vitro culturing conditions that simulate the 3D architecture of a tissue and/or organ. Therefore, this model has more correlation to the in vivo chemico-physical environment, which contributes to SC and neuron regulation, than the in vitro model based on just cell lines [86]. The main disadvantage, in comparison with the in vivo model of peripheral regeneration, is the limited time space provided for observation of regeneration. Cultures do not survive for months and resemble reconnection to target tissues. Therefore, these methods are fundamental to test a hypothesis and to be aware of the number of animals used but are not enough to have a fundamental experiment [86].

### 5.2. Animal Models

In PNI research, diverse animal models are commonly used. Rodents, mainly rats, are most frequently used as models of nerve injury following surgical repair or other treatments in study to assess the regeneration potency [55]. The choice of animal model depends on many factors, such as availability, easy care, specialized team to perform all the necessary procedures, lowest risk of infection, and tolerance to manipulation [45]. Another advantage is that functional outcomes like sensory, motor, and sympathetic recovery can be measured to demonstrate the outcome after PNI. To standardize the protocol, it is fundamental that the experimental injury, repair, and monitorization are carefully controlled and described. In addition, these models can provide more results, because at the end point, tissues can be harvested and analyzed. Normally, injured models involve damage of a nerve, followed by random animal aggrupation into test and control groups [55].

The use of rats as an animal model has some advantages, such as their resistance to infections and surgery complications, the availability of different strains, the well-studied morphological anatomy and low maintenance costs. Nevertheless, the small size of the rat can also be a limitation because the studies are performed in nerve gaps <10 mm. However, in humans the damage commonly exceeds 10 mm and nerve regeneration is slower [5,45]. Rodents’ metabolic rate is higher, in comparison with humans, so nerve regeneration is also faster which makes it difficult to understand the true effectiveness of the used therapies enhancing nerve repair [5,87,88]. These rodents’ extremely high neuroregenerative ability makes it harder to correlate to the human species [5,45].

To overcome the nerve gap size limitation other larger animal models are used, such as rabbits, mini-pigs, dogs, cats, and sheep. The rabbit model can be used to study larger nerve gaps (>50 mm), but there are more anatomical changes between these species and the human. Dogs and cats can be used, but the ethical issues regarding the use of this species has gradually reduced their use [5,45]. The sheep is commonly used due to the similarity between limb nerves [45]. Large models are the only solution to replicate the regeneration and effects of injuries in large nerve defects. Recently, an alternative model using the sheep was established. In this model, neurotmesis and axonotmesis injuries in the common peroneal nerve were induced. Afterwards, different therapeutic methods were applied to standardize the use of the sheep in nerve regeneration studies. To evaluate the nerve regeneration, a neurological examination protocol was established and adapted to the common peroneal nerve [88].

The zebrafish is another popular model that is used in several fields of regenerative medicine. This animal has a long superficial peripheral nerve in the posterior lateral line, and this line possesses the common cell types present in the PNS. Laser-mediated nerve transection coupled to time-lapse confocal microscopy is an advanced methodology that helps to investigate the degeneration and regeneration process of peripheral nerves in the zebrafish [20].

## 6. Quantifying Regeneration Following Peripheral Nerve Injury

### Methods to Measure Nerve Recovery in Animals

Commonly with PNI, gait changes occur. The previously described animal model is of utmost importance since the details of animal limb movements during locomotion help in the PNI assessment [23]. Various functional tests can aid in the motor recovery evaluation after PNI. These methods are fundamental to understand the therapeutic potential of some therapies [45]. As explained throughout this review, it has not yet been possible to select a therapeutic approach that has unequivocal efficacy. With these methods the optimal treatment, for each type of nerve injury, can be optimized and supported by scientific results. The methods are chosen depending on the study, cost, time to conduct the experience, and technical difficulty.

Concerning motor recovery, the majority of the tests are noninvasive and simple. They can be executed and help the monitoring throughout the recovery period as they are very important to fully understand nerve regeneration [55]. One of the methods to measure motor recovery using a functional index score, is the walking track/sciatic functional index (SFI) [23]. In this technique, the paw placement is monitored and also recorded to evaluate the animals’ walking, throughout a narrow corridor. It is a non-invasive method and provides access to the functional recovery of the hindlimb by recording and observing the footprint, considering the hindlimb as a whole. To perform this evaluation method the animal is placed inside a transparent acrylic corridor and bellow there is an image capture installed. The animal is placed in the beginning of the tunnel and to stimulate the animal to walk through the corridor there is a shelter in the end to encourage the animal to walk in that direction (140). Different parameters are analyzed so that it is possible to obtain the SFI [45,55]. An SFI of 0 is normal and an SFI of 100 indicates total impairment of the sciatic nerve [45]. The main limitations are the dragging of the tail during the footprint record, the development of flexion contracture, and smearing of the paws [5]. Basso, Beattie, and Bresnaha is a classic behavior method, which consists in the observation of a paralyzed rodent for 4 min in an open area. It is based on the assessment of some functional behaviors, like the paw placement and the limb movement. After, the locomotion the method allows to achieve a score from 0 (no observable locomotor movements) to 21 points (normal locomotor movements) [5,23,45].

The withdrawal reflex latency (WRL) is used to determine the integrity of nociceptive function. The animal is wrapped in a soft sloth and suspended over a heating plate at 56° temperature. WRL is defined by the time, in seconds, that the animal requires to retract its hind limb after making contact with the plate [61].

Another assessment, called static sciatic index (SSI), is obtained by the foot muscle function analysis. Which is basically performed with the animal in stationary position, to measure the toe spread [45]. The animal is placed inside a transparent acrylic box, and then placed inside the corridor. With both limbs in contact with the box floor this moment is caught in a camera below the acrylic box [61]. One innovative method is the use of an automated gait analysis system, CatWalk XT. This consists in a glass walking platform, recorded by a high-speed digital camera, illuminated by green light and the software is recording, analyzing, and processing the data collected from rodent paw prints. This method can overcome the human error of observation, because it uses specialized software to objectively measure several factors of gait. With this system it is easier to understand subtle gait changes and offers more parameters (associated to individual paw prints, paw prints position and coordination) to analyze [23]. Other described technique used animals walking on an acrylic track with 120 cm length, 12 cm width, and 15 cm height. Following trichotomy three reflective markers with 2 mm diameter were allocated to three different bony prominences on the desired hind limb. Then, a Visual 3D software provided the necessary measurements to determine the animal’s ankle angle [61]. Specially in PNIs, these methods can provide information about the upper and lower extremities, such as axonotmesis, neurotmesis, or fibrosis [23].

The whisker motion recovery after facial nerve injury, or muscle strength recovery after limb nerve recovery consists in recording animals for 3–5 min. Then, analyses of the angle, protraction, amplitude of retraction, and whisking frequency are evaluated, so it is considered a subjective evaluation [23,45,55].

The von Frey test is performed with the animal placed on a grid and the plantar skin is stimulated, by a noxious mechanical stimulus, using filaments. The purpose is to increase force until the animal responds by paw withdrawal. For comparison, both hind limb nerves are bilaterally evaluated at injured and noninjured sites [45,55]. There are other methods, such as the hot and cold response and the grasping test, that evaluate sensory function [55].

The number of animals can be reduced, if multiple non or minimally invasive functional tests are performed and compared to postmortem assessments. These techniques to assess axonal regeneration include histological analysis of the harvested tissue, immunohistochemistry, and retrograde/anterograde tracing [55]. Normally, histology is used to understand the number of axons in a repaired nerve. The different stains provides various types of information about the nerve regeneration, including the myelin, collagen fibers, cells, and non-neuronal cells such as SCs and macrophages [45,55]. Histological evaluation is more complex when compared to the classical staining, which makes it possible to perform quantitative and morphometric analyses of the histological sections. Toluidine blue staining semithin sections allow the identification of most myelinated axons and allow a good morphometric analysis. This staining helps to obtain parameters such as number, density and diameter of nerve fibers, perimeter and cross-section of fibers and axons, myelin sheaths and their thickness, and different ratios between the axon diameter [5]. Another useful measure, indicative of the innervation degree, is provided by the muscle distal to the nerve, that can be collected and weighed [55]. Furthermore, these results can be compared with the contralateral healthy side.

Neurophysiology is the stimulation of the nerve using electrical pulses [45]. To obtain these values, the electrodes are allocated proximal to the injury, under anesthesia. There are two possible options, the nerve and muscles can be exposed, and the electrodes allocated directly or, less invasively, with the aid of percutaneous needle electrodes [45,55]. The amplitude is related to the quantity of functionally reinnervated muscle nerve fibers and to estimate axonal loss. As an alternative, motor unit number estimation is another potential method to quantify the number of motor axons of the muscle being studied. To provide even more detailed values, compound nerve action potential—which means the record between two positions of the nerve—gives information about conduction in that specific segment under study [45,55].

## 7. Conclusions

PNI is a condition that affects the quality of life of many people and leads to motor and sensory deficits [19]. The positive point is that the PNS has some regeneration capacity, and the minor lesions can heal with full, or almost full, functional recovery in some cases[16]. However, as explained in this review, there are more severe lesions that frequently yield unsatisfactory outcomes, even with the gold standard methods of treatment. These severe injuries require treatment, mainly surgery, to help reconnecting the viable extremities of the nerve. In addition, when the nerve gap is bigger, there are other options, such as nerve grafts and NGC. Recently, there have been a lot of ongoing investigations and efforts to develop new treatment approaches, as new surgical repairs, next generation NGCs, and other techniques, including nutritional therapy, gene therapy, and the use of regenerative medicine. There are plenty of different MSCs that can enhance peripheral nerve regeneration, as well as their secretome. Even though these efforts to identify treatments are promising, it is very difficult to achieve a perfect technique that helps diminish neuropathic pain and helps in recovering the complete function of the nerve.

The in vitro and in vivo models were explained, as well as the methods to measure nerve recovery in animals, which are fundamental to understand the true meaning of a study concerning a new approach in the PNI. Besides, in a One Health approach, the translation of these models is crucial, even though there are some limitations. In summary, there are several promising new treatments and therapeutic approaches that with more studies will provide better outcomes for PNI in a near future, for both humans and animals. For that reason, this review is really important to revise the most common treatments and the recent ones, as well as to enhance the necessity to achieve a better gold standard that can assure better outcomes in almost all PNI severity lesions. It is necessary to continue investigating how to determine the ideal therapy for PNI by establishing standardized injury models and proper regenerative assessment methods.

## Figures and Tables

**Figure 1 ijms-23-00918-f001:**
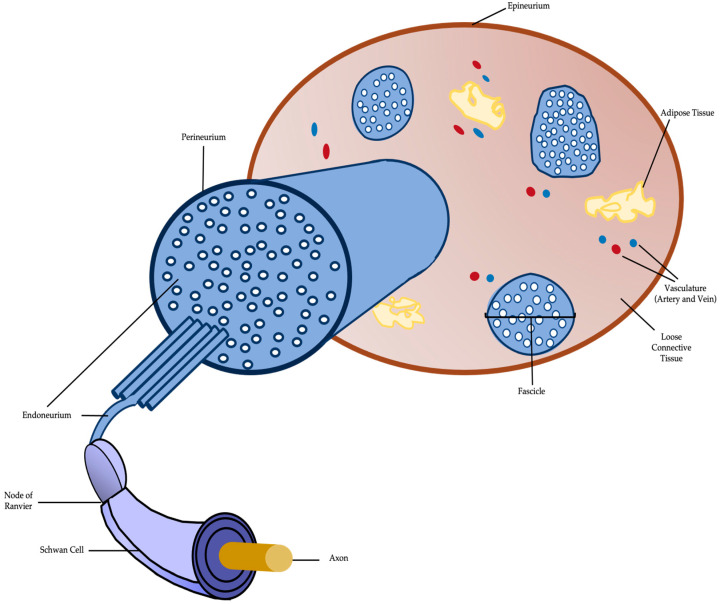
Schematic representation of a typical peripheral nerve.

**Figure 2 ijms-23-00918-f002:**
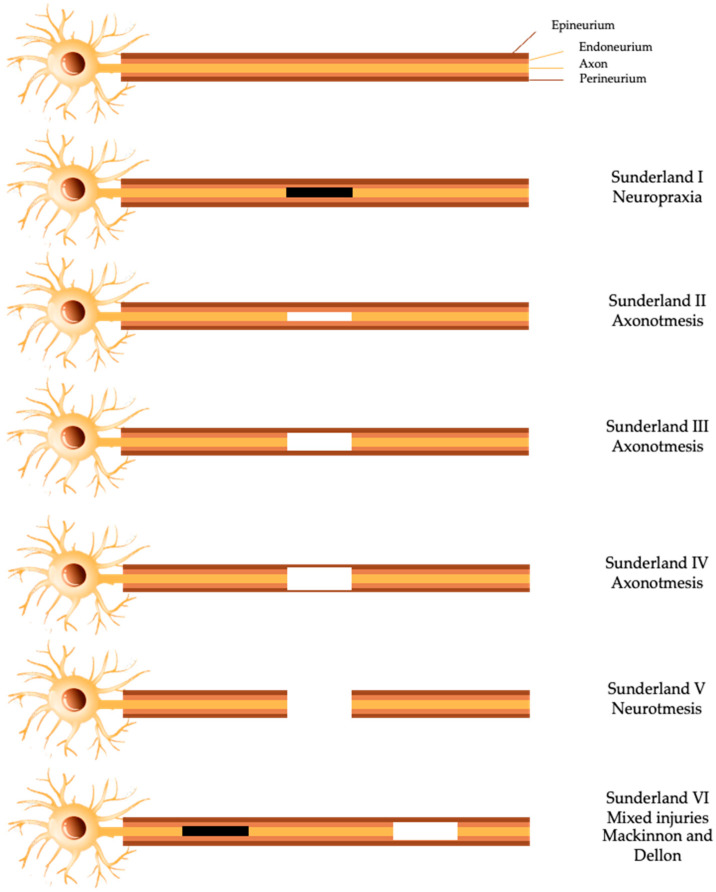
Schematic representation of the different injury grading systems for PNI.

**Figure 3 ijms-23-00918-f003:**
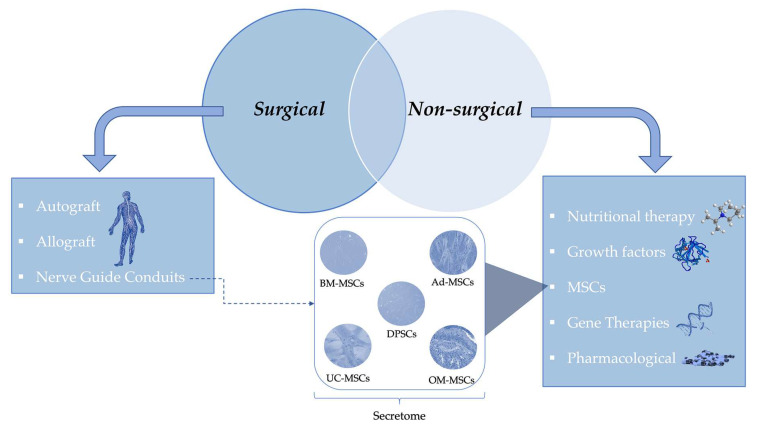
Schematic representation of the surgical and non-surgical treatments for PNI. The dotted arrow refers to the combination of nerve guide conduits with MSCs or their secretome, used as another possible therapeutic treatment for PNI. (Abbreviations: MSCs—Mesenchymal Stem Cells; BM-MSCs—Bone Marrow Mesenchymal Stem Cells; Ad-MSCs—Adipose-Derived Mesenchymal Stem Cells; DPSCs—Dental Pulp Stem Cells; UC-MSCs—Umbilical Cord Mesenchymal Stem Cells; OM-MSCs—Olfactory Mucosa Mesenchymal Stem Cells).

## Data Availability

Not applicable.

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
