# Peer review of "Peripheral Nerve Injury Treatments and Advances: One Health Perspective"

_ijms, 2022, doi:10.3390/ijms23020918_

Round 1

Reviewer 1 Report

The article entitled “Peripheral nerve injury treatments and advances: One Health perspective”. The aim of this review is to address the PNI and focus on the treatments and advances  achieved in peripheral nerve regeneration, especially using mesenchymal stem cells and biomaterialsis.

Below are some suggestions:

In the Abstract:

- Due to the importance of the repair process of peripheral nerve injuries, the authors could insert a paragraph emphasizing the importance of new techniques and therapeutic approaches already here in the abstract, thus demonstrating the relevance of the research.

 1.Introduction:

- One suggestion would be to remove the first paragraph that provides a general explanation of the nervous system, information that is already known. I would start right by talking about the peripheral nervous system.

3.Type of lesion and scarring

- Improve the quality of figure 2.

4.Advances in the peripheral nerve treatment

- In item 4.1.2 (line 269): where it is written "gap is bigger", what would be a large gap within peripheral nerve injuries? It is an important point and I suggest explaining it better.

* 4.3. Non-surgical treatment methods

- At this moment, several non-surgical therapies for the treatment of nerve injuries are cited and reviewed, such as: growth factors, mesenchymal stem cells, dental pup stem cells, secretame. Finally, I suggest including and discussing heterologous fibrin sealants as they are widely used in the treatment of peripheral lesions and play an important role in the scientific community within the tissue regeneration process (in PubMed there are more than 50 studies involving heterologous fibrin sealants ). During the reading, no item containing information about these sealants was observed.

6.Quantifying regeneration following peripheral nerve injury

- One suggestion would be to include a brief description also about morphometric analysis.

7.Conclusions:

- The conclusion is well written, it could only be adjusted more affirmatively in relation to the research objective.

Author Response

Review Report – Reviewer 1

Dear reviewer 1:

We would like to thank the reviewer´s comments on the manuscript “Peripheral nerve injury treatments and advances: One Health perspective". The manuscript has been modified and improved according to the suggestions of the reviewers. All changes introduced appear in the final document highlighted in yellow.

In the Abstract:

- Due to the importance of the repair process of peripheral nerve injuries, the authors could insert a paragraph emphasizing the importance of new techniques and therapeutic approaches already here in the abstract, thus demonstrating the relevance of the research.

As suggested by the reviewer, a paragraph was added with the relevance of this work (Line:  26-29).

“This review aims to enhance the importance of new therapies, especially in the severe lesions, to overcome limitations and achieve better outcomes. The urge for new approaches and the understanding of the different methods to evaluate nerve regeneration is fundamental in a One Health perspective.”

 1.Introduction:

- One suggestion would be to remove the first paragraph that provides a general explanation of the nervous system, information that is already known. I would start right by talking about the peripheral nervous system.

The following text segment was erased from the document (Line 35-42 and Line 46-48) to start the review talking about the peripheral nervous system.

“The coordination and communication throughout the body mainly depends on the highly specialized nervous system, which gives the possibility to respond to different environmental changes [1]. This appropriate response is generated when the central nervous system (CNS) obtains information from the receptors sent via afferent neurons.  Then, the CNS processes the information and the efferent neurons send signals to the different effector organs of the body [1]. The nervous system is divided in two major regions: CNS and peripheral nervous system (PNS) [2]. The brain and spinal cord constitute the CNS, which is responsible for centrally process the information arriving from the periphery [1].”

“Neuropathic pain is one of the most complicated pain syndromes to regulate and can cause this kind of disorder can cause devastating impacts on patients’ daily routines and impair their emotional well-being [6, 7].”

3.Type of lesion and scarring

- Improve the quality of figure 2.

The quality of figure 2 was improved.

4.Advances in the peripheral nerve treatment

- In item 4.1.2 (line 269): where it is written "gap is bigger", what would be a large gap within peripheral nerve injuries? It is an important point and I suggest explaining it better.

The following text segment has been introduced into the document in order to explain what is considered a large gap (Line:  267-268).

“is above the critical size, which is about 3 cm in humans, considered a critical nerve injury…”

* 4.3. Non-surgical treatment methods

- At this moment, several non-surgical therapies for the treatment of nerve injuries are cited and reviewed, such as: growth factors, mesenchymal stem cells, dental pup stem cells, secretome. Finally, I suggest including and discussing heterologous fibrin sealants as they are widely used in the treatment of peripheral lesions and play an important role in the scientific community within the tissue regeneration process (in PubMed there are more than 50 studies involving heterologous fibrin sealants). During the reading, no item containing information about these sealants was observed.

The authors agree with the Reviewer comment and propose the following to be introduced in the manuscript (Line 221-236 and Line 810-826):

“Another strategy to reduce the number of stiches for reconstruction after PNI is the use of suture associated with heterologous fibrin sealant (HFS) [25]. HFS contains a fibrinogen-rich cryoprecipitate extracted from a buffalo’s blood (Bubalus bubalis) in addition to a thrombin-like enzyme purified from snake venom (Crotalus durissus) [25, 26]. Furthermore, HFS is important to the healing process because the combination of fibrin with the proteins enhances angiogenesis, wound contraction, collagen synthesis and re-epithelization. The use of sutures can lead to inflammatory reactions such as granuloma and neuroma formation, so reduce the number of points used in the reconstruction is a way to minimize the damage and the inflammatory responses [24]. Leite et al., compared the use of HFS associated with a reduced number of stiches, 1 point, in comparison to the conventional suture, 3 points, after ischiatic nerve injury in the rat model [24]. To prove the regeneration benefits of using HFS the group used some regeneration evaluations such as Catwalk, electromyography and morphometric analyses. The results showed that HSF sealant adjuvant to the suture had superior values concerning axons and nerve fibers diameter and area and better muscle weight. Therefore, these results suggest a protective effect of HSF and a decrease of trauma due to the stiches reduction [24].”

“One study used the rat model after sciatic nerve neurorrhaphy and tested the combination of human embryonic stem cells, genetically modified to overexpress basic fibroblast growth factor, with HFS scaffold. The aim was to enhance regeneration and neuronal survival using autografting with end-to-end neurorrhaphy [82]. To determine if the use of this combination was a good solution for PNI, the authors used the walking track analysis, immunohistochemistry and the von-Frey test. The results proved that the application of this scaffold incorporated with human embryonic stem cells, on the site of injury, successfully regenerated sensory and motor fibers [82].

Other groups used a fibrin sealant (FS) derived from venom of the snake Crotalus durissus terrificus, as support for MSCs to enhance regeneration [83, 84]. Spejo et al., applied a FS scaffold incorporated with BM-MSCs in filling the gap formed by the induced lesion at the spinal levels (L4, L5 and L6) in the rat model [83]. The results demonstrated that this combination caused greater expression of anti-inflammatory cytokine IL-1ß and proinflammatory M1 macrophages. However, the environment provided by the scaffold could have altered the BM-MSCs properties guiding to a proinflammatory milieu that resulted in the absence of motor improvement, in comparison with the group only using MSCs as a treatment [83]. “

  1. Leite, A.P.S., et al., Heterologous fibrin sealant potentiates axonal regeneration after peripheral nerve injury with reduction in the number of suture points. Injury, 2019. 50(4): p. 834-847.

  1. Abbade, L.P.F., et al., Treatment of Chronic Venous Ulcers With Heterologous Fibrin Sealant: A Phase I/II Clinical Trial. Front Immunol, 2021. 12: p. 627541.

  1. Ferreira, R.S., Jr., et al., Heterologous fibrin sealant derived from snake venom: from bench to bedside - an overview. J Venom Anim Toxins Incl Trop Dis, 2017. 23: p. 21.

  1. Mozafari, R., et al., Combination of heterologous fibrin sealant and bioengineered human embryonic stem cells to improve regeneration following autogenous sciatic nerve grafting repair. J Venom Anim Toxins Incl Trop Dis, 2018. 24: p. 11.

  1. Spejo, A.B., et al., Neuroprotection and immunomodulation following intraspinal axotomy of motoneurons by treatment with adult mesenchymal stem cells. J Neuroinflammation, 2018. 15(1): p. 230.

  1. Frauz, K., et al., Transected Tendon Treated with a New Fibrin Sealant Alone or Associated with Adipose-Derived Stem Cells. Cells, 2019. 8(1).

And the following abbreviations were introduced in the document (Line 1073 and Line 1078):

FS – Fibrin sealant

HFS – Heterologous fibrin sealant

6.Quantifying regeneration following peripheral nerve injury

- One suggestion would be to include a brief description also about morphometric analysis.

As suggested by the reviewer, a paragraph was added with a description about morphometric analysis (Line 994-1000).

“Histological evaluation is more complex when compared to the classical staining, which makes it possible to perform quantitative and morphometric analyses of the histological sections. Toluidine blue staining semithin sections allows the identification of most myelinated axons and allows a good morphometric analysis. This staining helps to obtain parameters such as number, density and diameter of nerve fibers, perimeter and cross-section of fibers and axons, myelin sheaths and its thickness and different ratios between the axon diameter [5].”

7.Conclusions:

- The conclusion is well written, it could only be adjusted more affirmatively in relation to the research objective.

According to the reviewer's comment, the conclusion was modified in order to highlight the objective and relevance of the presented article. The following text segment was introduced in the document (Line 1032-1039).

“In summary, there are several promising new treatments and therapeutic approaches that with more studies will provide better outcomes for PNI in a near future, for both humans and animals. For that reason, this review is really important to revise the most common treatments and the recent ones, as well as, to enhance the necessity to achieve a better gold standard that can assure better outcomes in almost all PNI severity lesions. It is necessary to continue investigating to determine the ideal therapy for PNI by establishing standardized injury models and proper regenerative assessment methods.”

Reviewer 2 Report

A very interesting overview about the possible therapeutic strategies in peripheral nerves injuries (surgery, pharmacological therapies) with possible suggestions about new strategies, such as cell-based therapies or biomaterials. The authors also provide a preclinical explanation of the efficacy of such strategies, from a translational point of view. However, whereas the concept of One Health is provided yet in the title, I think that such an approach should be better detailed. In other words, the pathophysiology and anatomy of peripheral nerves and related injuries are really well-known concepts, and I think that deserving to these topics too much space within the manuscript reduce the readers' experiences (the first 5 pages and related figures should be greatly reduced because this structure appears more similar to a chapter than to an article). On the other hand, figures supporting the therapeutic and translational approaches can greatly increase the manuscript value, which is quite long. For example, it could be possible to insert a scheme or figure explaining the whole possible therapeutic strategies? with the underlying rationale from a preclinical point of view.

Author Response

Review Report - Reviewer 2

Dear reviewer 2:

We would like to thank the reviewer´s comments on the manuscript “Peripheral nerve injury treatments and advances: One Health perspective". The manuscript has been modified and improved according to the suggestions of the reviewers. All changes introduced appear in the final document highlighted in yellow.

A very interesting overview about the possible therapeutic strategies in peripheral nerves injuries (surgery, pharmacological therapies) with possible suggestions about new strategies, such as cell-based therapies or biomaterials. The authors also provide a preclinical explanation of the efficacy of such strategies, from a translational point of view. However, whereas the concept of One Health is provided yet in the title, I think that such an approach should be better detailed. In other words, the pathophysiology and anatomy of peripheral nerves and related injuries are really well-known concepts, and I think that deserving to these topics too much space within the manuscript reduce the readers' experiences (the first 5 pages and related figures should be greatly reduced because this structure appears more similar to a chapter than to an article). On the other hand, figures supporting the therapeutic and translational approaches can greatly increase the manuscript value, which is quite long. For example, it could be possible to insert a scheme or figure explaining the whole possible therapeutic strategies? with the underlying rationale from a preclinical point of view.

The authors agree with the Reviewer comment and propose the following to detail the One Health approach, page 2, line 80-91:

“Human and veterinary medicine can mutually benefit from research that applies a one Health perspective [14]. The enforcement of transdisciplinary strategies can improve the knowledge about the well-being of animals, humans and the environment [15]. In PNI the use of animals as models help to achieve more results concerning different treatments because clinical data from humans can have economic, practical or ethical limitations. For that reason, the correct use of animals has proven value for later human clinical trials in PNI, that happens both in humans and animals. For that purpose, it is important to choose correctly the best animal model to use in each study to draw meaningful conclusions from animal experiments and translate to human medicine. The One Health perspective in PNI benefits both human and veterinary medicine in the urge to obtain a treatment that enhances the outcome from PNI, especially in severe lesions. “

  1. Ribitsch, I., et al., Large Animal Models in Regenerative Medicine and Tissue Engineering: To Do or Not to Do. Front Bioeng Biotechnol, 2020. 8: p. 972.

  1. Franco-Martinez, L., et al., Biomarkers of health and welfare: A One Health perspective from the laboratory side. Res Vet Sci, 2020. 128: p. 299-307.

The following text segment was erased from the document (Line 35-42, Line 46-48, Line 74-76, Line 97-99, Line 106-109, Line 126-136, Line 159-160, Line 165-167, Line 169-172) to reduce the space occupied by some well-known concepts.

The coordination and communication throughout the body mainly depends on the highly specialized nervous system, which gives the possibility to respond to different environmental changes [1]. This appropriate response is generated when the central nervous system (CNS) obtains information from the receptors sent via afferent neurons.  Then, the CNS processes the information and the efferent neurons send signals to the different effector organs of the body [1]. The nervous system is divided in two major regions: CNS and peripheral nervous system (PNS) [2]. The brain and spinal cord constitute the CNS, which is responsible for centrally process the information arriving from the periphery [1].”

“Neuropathic pain is one of the most complicated pain syndromes to regulate and can cause this kind of disorder can cause devastating impacts on patients’ daily routines and impair their emotional well-being [6, 7].”

“These have many advantages, in comparison with autologous nerve graft, such as unrestricted source of supplement, easy fabrication, repeatability and prevention of donor site morbidity”

“In addition, depending on the size and the specific type of nerve, it represents between 30-70% of the nerve trunk sectional area [5].”

“In comparison with the perineurium and epinerium, which are circumferentially disposed, the endonerium has a longitudinal orientation [5]. The nerve fascicle is a group of axons surrounded by endonerium and each fascicle is covered by perinerium [5]. “

“Along the myelinated fiber there is a heterogenous distribution of sodium channels, these channels are in bigger density at the Ranvier nodes and at a minor density in the internodal region. The inner regions lose small quantities of sodium to the extracellular fluid, which helps to maintain a separation of electrical charges, in opposition to the Ranvier nodes where the sodium channels density is higher. When sodium is in the extracellular fluid, a minimum quantity is enough to open the sodium channels, the ion penetrates the axon causing the action potential [5]. The action potential travels fast along the myelinated axon by jumping between the nodes of Ranvier. For axons of the same size, myelination accelerates nerve conduction in 20-100-fold, in comparison with unmyelinated axons, which is the reason why, normally, unmyelinated axons are smaller [7].”

“depending on the recovery time, prognosis and severity of nerve injury [7, 16, 17]

“Although being a more severe lesion, the injured nerve stump maintains some integrity of the physiological structure and organization, and functional recovery is expected [16]

“because the disconnection of the two segments damages the collagen coatings and their guiding function implicating the normal regenerative sequence.”

“There is the possibility to have lesions with different severities throughout the nerve’s cross section and extension “

The following text segment was introduced in the document (Line 855-857)

“To summarize this chapter about the available treatments for PNI, Figure 3 divides this section in two major options, surgical and non-surgical, showing the different possibilities in each one.”

And the following figure (Figure 3) was introduced in the document to summarize the different possible therapeutic approaches.

Figure 3: Schematic representation of the surgical and non-surgical treatments for PNI (Abbreviations: MSCs - Mesenchymal Stem Cells; BM-MSCs - Bone Marrow Mesenchymal Stem Cells; Ad-MSCs - Adipose-Derived Mesenchymal Stem Cells; DPSCs – Dental Pulp Stem Cells; UC-MSCs – Umbilical Cord Mesenchymal Stem Cells; OM-MSCs - Olfactory Mucosa Mesenchymal Stem Cells).

Round 2

Reviewer 1 Report

All suggestions presented were carried out. Thanks!

Reviewer 2 Report

The manuscript in the present form has greatly increased its scientific value, and I think that it is now suitable for publication